# Clinical and Epidemiological Characteristics of Bloodstream Infections in Head and Neck Cancer Patients: A Decadal Observational Study

**DOI:** 10.3390/jcm11164820

**Published:** 2022-08-17

**Authors:** Shiori Kitaya, Risako Kakuta, Hajime Kanamori, Akira Ohkoshi, Ryo Ishii, Kazuhiro Nomura, Koichi Tokuda, Yukio Katori

**Affiliations:** 1Department of Otolaryngology, Head and Neck Surgery, Tohoku University Hospital, Sendai 980-8574, Japan; 2Department of Infectious Diseases, Internal Medicine, Tohoku University Hospital, Sendai 980-8574, Japan

**Keywords:** blood cultures, head and neck cancers, bloodstream infections, catheter-related bloodstream infections

## Abstract

This retrospective study aims to describe the clinico-epidemiological characteristics of bloodstream infections (BSIs) and the risk factors in patients with head and neck cancer (*n* = 227) treated at the Department of Otolaryngology, Head and Neck Surgery, Tohoku University Hospital between April 2011 and March 2021. Overall, 23.3% of blood cultures were positive. In the culture-positive group, catheter-related bloodstream infections (CRBSIs) were the most common (38.8%), followed by respiratory tract infections (19.4%), and catheter-associated urinary tract infections (6.0%). Methicillin-resistant *Staphylococcus aureus* (26.9%), *Staphylococcus epidermidis* (17.9%), and *Pseudomonas aeruginosa* (10.4%) infections were common. The most frequent treatment for head and neck cancer was surgery (23.9%), followed by treatment interval or palliative care (19.4%), and single radiotherapy (13.4%). The 30-day mortality rate was significantly higher in the BSI than in the non-BSI group (10.4% vs. 1.8%, respectively). CRBSIs are the most frequent source of BSIs in patients with head and neck cancer. In conclusion, central venous catheters or port insertion should be used for a short period to prevent CRBSIs. The risk of developing BSI should be considered in patients with pneumonia. Understanding the epidemiology of BSIs is crucial for diagnosing, preventing, and controlling infections in patients with head and neck cancer.

## 1. Introduction

Because of decreased basal immunity, frequent invasive procedures, and antineoplastic chemotherapy, patients with solid tumors are at high risk of bloodstream infection (BSI). It is estimated that 5.5–16.4% of such patients develop BSIs [1]. Head and neck cancers are the sixth most common malignancy worldwide, with approximately 600,000 newly diagnosed cases annually [2]. The treatment for head and neck cancers is broadly divided into two types: surgical and chemoradiotherapy. Reconstructive surgical treatment is often invasive; therefore, patients require long-term rehabilitation for exercise and swallowing. Myelosuppression is common in patients receiving chemoradiotherapy due to the side effects of anticancer drugs, sometimes leading to neutropenia [3] and impairment of the skin and pharyngeal mucosa due to the side effects of radiotherapy [4]. These side effects may require long-term treatment and hospitalization; thus, the risk of BSIs in patients with head and neck cancer, similar to other solid tumors and hematological malignancies, may be high. The epidemiological characteristics of BSIs in patients with hematological malignancies and solid tumors have been described [5,6]; however, the characteristics of BSIs in patients with head and neck cancer have not been described and remain largely unknown.

This study aims to describe the risk factors as well as the clinical and epidemiological characteristics of BSIs in patients with head and neck cancer.

## 2. Materials and Methods

### 2.1. Patients

Electronic charts (Tohoku University Hospital) of patients who underwent blood culture procedures during hospitalization or outpatient visits to the Department of Otolaryngology, Head and Neck Surgery—between April 2011 and March 2021—were screened. The data for each bloodstream isolate were collected from the infection-control computerized records. The following anamnestic and clinical data were obtained from the questionnaire and medical records: Sex, age, comorbidity, social history, vital signs, blood test results (levels of serum white blood cells, neutrophils, and C-reactive protein), presence of neutropenia, mortality, duration of hospitalization, use of antibiotics, participation of the infectious disease department in the treatment, sets of blood cultures, infection variables (associated sites of infection, organisms, and susceptibility), presence of a central venous catheter (CVC)/central venous port (CV port) and its indwelling period, primary tumor site, and the types of treatment administered based on blood culture reports. The data on microorganisms were extracted from the database of the infectious disease department. The Human Ethical and Clinical Trial Committee of Tohoku University Hospital approved this survey (2018-1-736).

### 2.2. Definitions and Outcomes

Blood cultures were obtained when an infection was suspected based on vital signs (e.g., presence of fever, tachycardia, hypotension, and shock), clinical findings (e.g., productive cough, pain on urination, or abdominal pain), laboratory findings (e.g., levels of serum white blood cells, neutrophils, and C-reactive protein, and presentation as organizing pneumonia on chest radiography), and medical history (e.g., recent surgery, chemoradiotherapy). Blood samples were obtained from all patients when BSIs were suspected, and the effects of serum white blood cells, neutrophils, and C-reactive protein levels on the clinical findings were determined. Cultures of other sites were obtained, depending on the source of infection, when signs of localized infection were suspected (e.g., sputum, urine, and pus culture). The sources of bacteremia were determined from medical records, clinical findings, imaging studies, and microbiological evidence from otolaryngologists and infectious disease physicians. Patients receiving insulin injections were defined as having diabetes mellitus.

We classified the patients into two groups, i.e., the BSI or non-BSI group. A positive BSI diagnosis was defined as when one or more blood cultures were positive for known pathogenic organisms (e.g., *Staphylococcus aureus*, gram-negative bacilli, and fungi). Patients who were not in the BSI group were considered as belonging to the non-BSI group. Contamination cases were included in the non-BSI group. Contamination was estimated if the blood cultures were positive for *Corynebacterium* spp., *Bacillus* spp., or *Propionibacterium acnes*. Cases in which coagulase-negative staphylococci (CoNS) were isolated from one of the two sets of blood cultures were also regarded as contamination cases [7]. If the same pathogen grew in multiple positive blood cultures, a bacteremic episode was considered. All pathogens were considered separately if more than one microorganism was isolated from a single blood culture. Polymicrobial infection was defined as an infection in which more than one species of the pathogen was isolated in a single or separate blood culture specimen within the same BSI episode. For patients who had undergone multiple episodes of blood cultures, multiple findings of bacteremia were considered different when negative blood cultures were observed between episodes and in cases where there were different clinical findings or microorganisms.

Catheter-related BSIs (CRBSIs) were defined according to the guidelines of the Infectious Diseases Society of America [8], and include any of the following conditions: Identification of the same organism from at least one percutaneous blood culture and catheter tip; two blood cultures (one from the catheter hub and one from the peripheral vein) that meet the CRBSI criteria for quantitative culture or differential time to positivity; or two quantitative blood cultures from two catheter lumen cultures where the colony count for blood drawn from one lumen is three-fold greater than that of blood drawn from the second lumen. All CVC or CV port insertions were performed in accordance with standard protocols at the CV center; thus, the procedure was performed under fluoroscopy with maximal sterile barrier precautions, including the use of a cap, mask, sterile gown, sterile gloves, and a sterile full-body drape. Basically, the CVC or CV port was removed and cultured when the possibility of CRBSI was suspected [8]. If the infecting organism was identified as CoNS and there was no suspicion of local or metastatic complications, the CVC or CV port was retained. Neutropenia was defined as an absolute neutrophil count of <500/mm^3^ when BSIs occurred, consistent with the common terminology criteria for adverse events, version 5.0 (CTCAE v5.0) [9]. Antimicrobial therapy was considered inappropriate when at least one of the following conditions were met: An administration of ineffective antimicrobial agents, which were provided based on the antimicrobial susceptibilities in the organisms identified in the blood culture; continued use of initial antibacterial drugs despite the de-escalation of cases and declaration of pathogens from blood cultures and their sensitivity [10]; when treatment duration was not in line with current medical standards [11]. Antimicrobial therapy was considered appropriate when no criterion that would satisfy the definition of inappropriate use was met. Histological type and grading of the tumor were evaluated according to the standard International Classification of Diseases for Oncology third edition [12], third edition first revision [13], and third edition second revision [14]. The tumors were staged according to the UICC TNM classification (sixth edition UICC 2002 [15], seventh edition UICC 2010 [16], and eighth edition UICC 2017 [17]). In terms of the head and neck cancer treatment group classification, the following parameters were included in each treatment group: Onset within 30 days after surgery, various chemotherapies, and radiation therapy. Subsequent onsets were included in the treatment interval group (e.g., a case of blood culture collection on postoperative day 45). In our hospital, all chemoradiotherapy procedures were performed during hospitalization. Moreover, 30-day mortality was considered as death due to an infectious disease.

The primary outcome variable in this study was the 30-day mortality rate. The secondary outcome variable was hospitalization duration.

### 2.3. Method for Collecting Blood Cultures

During the study period, arterial or venous blood was aseptically obtained and inoculated into aerobic and anaerobic BACT/ALERT^®^ FA plus bottles (BioMérieux, Durham, NC, USA) for patients with suspected BSIs. Each bottle was incubated in a BACT/ALERT VIRTUO instrument (BioMérieux, Durham, NC, USA) at 37.0 °C for 7 days. Identification and susceptibility tests were performed using a VITEK MS system (BioMérieux, Métropole de Lyon, France) and Walk Away 96 Plus system (Siemens Healthcare Diagnostics, Deerfield, IL, USA), respectively. As for fungi and *Haemophilus influenzae*, the susceptibility tests were performed using a RAISUS S4 (Nissui Pharma, Tokyo, Japan).

### 2.4. Statistical Analysis

We performed a Mann–Whitney U test to compare the averages of continuous variables (e.g., age, body weight, and body temperature) and Fisher’s exact test was used to compare proportions of categorical variables (e.g., sex and presence of medical history) between the BSI and non-BSI groups. A Kaplan–Meier survival analysis was performed to test survival differences between patients with and without BSIs. The means were compared using the log-rank test. We performed a multivariate logistic regression analysis to identify the risk factors for BSI and 30-day mortality. Variables with a *p*-value <0.10 in the univariable analysis were examined for correlation before inclusion in the multivariate analysis. The analysis was performed using SPSS version 27 (IBM Corp., Armonk, NY, USA). Differences were considered statistically significant at a corrected *p*-value <0.05.

## 3. Results

The characteristics of BSIs in patients with head and neck cancer are presented in Table 1. During the study period, a total of 4958 patients with head and neck cancer (excluding duplicate patients) were hospitalized or underwent relevant outpatient surgery. Blood cultures were collected from 288 (5.8%) of these patients. The BSI group had a large proportion of patients aged 60–80 years (men: 88.0%, women: 82.4%). There were no significant differences in demographic characteristics, comorbidities, social history, frequency of neutropenia, or hospitalization duration between the BSI and non-BSI groups. There was no change in the obvious time series. Interestingly, the 30-day mortality rate was significantly higher in the BSI than in the non-BSI group [seven cases (10.4%) vs. four cases (1.8%), *p* = 0.004]. The outcome of the Kaplan-Meier analysis of the entire patient cohort is presented in Figure 1. The Kaplan-Meier survival estimates showed no significant difference over time between the BSI and non-BSI groups (*p* = 0.450). The reason for this finding in cases wherein antimicrobials were not used appropriately included the following: Insufficient treatment duration (four cases, 30.8%), extensive treatment duration (two cases, 15.4%); continued use of initial antimicrobials although de-escalation was possible (four cases, 30.8%); administration of ineffective antimicrobial agents (two cases, 15.4%); insufficient treatment duration and continued use of initial antimicrobials although de-escalation was possible (one case, 7.7%). The most frequent types of infection were CRBSI (twenty-six cases, 38.8%), followed by respiratory tract infection (thirteen cases, 19.4%) and catheter-associated urinary tract infection (four cases, 6.0%) in patients with BSI with head and neck cancer. The occurrence of CRBSI, pyogenic spondylitis, and thrombophlebitis was significantly higher in the BSI than in the non-BSI group (*p* < 0.001, *p* = 0.040, and *p* = 0.012, respectively). When comparing CRBSIs with other infectious diseases, there were no significant differences in hospitalization and 30-day mortality; however, the 30-day mortality rate tended to be lower in CRBSI than in non-CRBSI cases [one (3.8%) vs. six cases (14.6%)]. The presence of a CVC or CV port was significantly higher in the BSI than in the non-BSI group (*p* < 0.001 and *p* = 0.004, respectively). The ratios of CVC or CV port removal were not significantly different between the BSI and non-BSI groups. Concerning the BSI group, a catheter tip culture was performed in all cases where the catheter was removed. There were fifteen cases (65.2%) in which the same bacterial pathogens in the blood culture were detected from the catheter tip; there were five cases (21.7%) where the catheter tip culture was negative. Catheter removal followed the administration of antibacterial drugs and only after a considerable amount of time had passed in many cases, explaining why the bacteria were not detected in some catheter-tip cultures. There were three cases (13.0%) of suspected contamination. The most frequent treatment for head and neck cancer was surgery (sixteen cases, 23.9%), followed by treatment interval or palliative care (thirteen cases, 19.4%) and single radiotherapy (nine cases, 13.4%). There was no significant difference in the sites or stage of cancer and types of treatment between the BSI and non-BSI groups.

The results of the multivariate logistic regression analysis are presented in Table 2 and Table 3. Factors independently associated with BSI in patients with head and neck cancer were body temperature (adjusted odds ratio [aOR], 2.563; 95% CI, 1.829–3.593) and C-reactive protein level (aOR, 1.047; 95% CI, 1.009–1.085). Cisplatin radiation therapy (CDDP-RT) was an independent factor that reduced BSI prevalence (aOR, 0.336; 95% CI, 0.129–0.870).

The organisms detected from blood cultures of patients with head and neck cancer are presented in Table 4. Gram-positive and gram-negative organisms were isolated in 49 (73.1%) and 28 (41.8%) episodes, respectively. Fungal pathogens represented six (9.0%) episodes. Among the patients with head and neck cancer, methicillin-resistant *Staphylococcus aureus* (*n* = 18, 26.9%), *Staphylococcus epidermidis* (*n* = 12, 17.9%), *Pseudomonas aeruginosa* (*n* = 7, 10.4%), *Enterobacter aerogenes* (*n* = 6, 9.0%), *Klebsiella pneumoniae* (*n* = 4, 4.8%), and methicillin-sensitive *Staphylococcus aureus* (*n* = 4, 4.8%) were the most frequently recovered organisms from blood cultures. In CoNS, the methicillin resistance rate was 76.5% (13/17 cases). *Staphylococcus epidermidis* (*p* < 0.001) and *Candida parapsilosis* (*p* = 0.044) infections were significantly more prevalent in the CRBSI group than other infectious diseases.

## 4. Discussion

To our best knowledge, this is the largest study that has analyzed the clinical and epidemiological characteristics of blood cultures obtained from patients with head and neck cancer. We found that the most frequent source of BSIs in patients with head and neck cancer in our department was CRBSIs. There was no significant difference in mortality or average duration of hospitalization between the BSI and non-BSI groups.

A previous retrospective cohort study found a 20% infection mortality rate for patients with a BSI and solid tumors [5]. In a recent study of BSIs in patients with head and neck cancer, the 30-day mortality rate was 26%, and BSIs were involved in 10% of early non-cancer deaths [18]. In this study, the 30-day mortality rate of patients with head and neck cancer with BSI was 13.4%. The mortality rate in this study was slightly lower than those in previous studies. The 30-day mortality rate was significantly higher in the BSI group than in the non-BSI group. Therefore, it is considered that death due to BSI is likely to occur in the early stage of infection. Thus, early detection and treatment of BSI may be helpful in reducing mortality from BSI. A prospective cohort study also reported that the mean duration of hospitalization for patients with bacteremia with solid organ malignancies was 21.8 days [19]. In this study, the mean duration of hospitalization for patients with BSI with head and neck cancer was 86.0 days. The reason for the long hospitalization period was the need for either long-term rehabilitation after reconstructive treatment or an extended treatment period with chemoradiotherapy; this usually takes approximately 3 months but becomes longer if cervical dermatitis or pharyngitis occurs because of the side effects of radiotherapy. In previous studies on the significant risk factors for BSIs with head and neck cancer, Marin et al. reported hypoalbuminemia, previous chemotherapy, and cetuximab therapy, while Jensen et al. reported increasing age, stage, and performance status [18,20]. In this study, body temperature, C-reactive protein level, and CDDP-RT were significantly related to an increased risk in BSI development. Each study identified different related factors, but the factors are common and lead to exacerbations of overall ill condition and decreased activity.

Gram-positive bacteria were more commonly isolated in our study, which was similar to the results of a previous French multicenter prospective study of patients with cancer and bacteremia [21]. Alternatively, in other published prospective cohort studies, the results are conflicting; there is a higher incidence of gram-negative pathogens among patients with BSI and solid organ or hematologic malignancies [19]. Gram-positive bacteria have been reported to cause approximately 60% of documented BSIs in many institutions [22], and the prevalence of gram-positive organisms as causative microorganisms of BSIs has substantially increased recently [23]. The most commonly reported causative pathogens of CRBSIs are gram-positive bacteria, such as CoNS, *Staphylococcus aureus*, and enterococci, followed by *Candida* spp. [24]. One of the reasons for the common detection of gram-positive bacteria in this study may be that CRBSIs were the most frequent cause of BSIs in patients with head and neck cancer. Regarding antimicrobial therapy, cases of inappropriate treatment were more frequent when the infectious disease department did not participate in the treatment. There was no significant difference in mortality between the group of patients with BSI who received intervention from the infectious disease department and the group who did not receive any such intervention. However, the 30-day mortality rate tended to be lower in the group with intervention from the infectious disease department than in the group without any such intervention [three (7.0%) vs. four cases (16.7%)]. In our hospital, almost all cases of blood culture data collected are sent to the infectious disease department for diagnosis. This department has been advising antimicrobial therapy to other departments since 2012. It is desirable that the infectious disease department actively participates in the treatment of BSIs because the selected antibacterial agents and treatment periods vary depending on the causative organisms and infected organs [25]. Furthermore, the intervention of the infectious disease department may contribute to the proper use of antimicrobial agents and lead to favorable outcomes in the treatment of BSIs in patients with head and neck cancer. 

In this study, the most frequent source of BSIs in patients with head and neck cancer was CRBSIs, followed by pneumonia. In the reports of Marin et al., pneumonia was the most common focus of infection (fourteen cases, 27.0%), CRBSI was the third most common (eight cases, 16.0%) [20], and the results were roughly similar to those in this study. The results suggest that CRBSI and pneumonia are important as foci of BSI in patients with head and neck cancer. Patients with head and neck cancer often receive CVC or CV ports to administer intravenous medications, such as anticancer drugs, molecular-targeted drugs, and immune checkpoint inhibitors, when the placement of peripheral venous routes is difficult or when the aim of safety is antineoplastic chemotherapy or total parenteral nutrition [26]. Despite the benefits of CVCs and CV ports, they also have the potential risk of BSIs, especially CRBSIs [27]. Additionally, the risk of CRBSI increases as the duration of CVC or CV port insertion increases [28], given the treatment period extension or outpatient treatments. Due to these reasons, the incidence of CRBSIs in patients with head and neck cancer is also high, as it is in patients with other solid tumors and hematological malignancies. In our hospital, CVC and CV ports were inserted with maximal sterile barrier precautions; however, CRBSIs were the most common cause of BSIs. Therefore, it is important to manage the catheter not only at the time of insertion but also after insertion. Chlorhexidine alcohol is used for skin disinfection during catheter insertion and for changing dressing materials at a frequency that is suitable for preventing CRBSIs, considering the type of catheter. Recently, the use of chlorhexidine-impregnated dressings or antimicrobial/antiseptic impregnated catheters were shown to reduce the incidence of CRBSIs [29]. In addition, the use of taurolidine lock solution reduces the risk of CRBSIs, without obvious adverse effects or bacterial resistance [30]. It is desirable to remove the catheter when CRBSIs occur; however, catheter removal may be difficult in cases where the placement of peripheral venous routes is challenging. Antibiotic lock therapy using drugs, such as vancomycin and daptomycin, as well as ethanol lock therapy are options that may be particularly effective in such cases [31,32]. 

CRBSIs were the most frequent BSIs; however, the mortality rate attributed to them was significantly lower than that attributed to other infectious diseases in this study. This was due to the established guidelines for the treatment of CRBSIs and their relatively easy management, including the removal of the CVC or CV port and antimicrobial catheter lock, compared to that of other infectious diseases. Early therapeutic interventions may have contributed to the lower mortality rate in CRBSI cases [33]. In addition, as the Department of Infectious Diseases advises the attending physician on the method of treatment and duration of therapy for CRBSI, this intervention may also have led to a decrease in mortality. Decreasing the frequency of CRBSIs may reduce the duration of hospitalization and antibiotic therapy, which may prevent the advancement of drug-resistant bacteria [34]. CRBSIs are also one of the most expensive complications of central venous catheterization [30]; hence, a reduction in their frequency may lead to a decrease in clinical staff effort and laboratory volume, which may result in reduced medical expenses.

A retrospective study in Spain showed that pneumonia (25.6%) was the second most common cause of BSIs after the endogenous source (26.3%) in patients with solid tumors who presented with neutropenia [35]. Dysphagia is a major cause of chronic aspiration and aspiration pneumonia [36]. In patients with advanced head and neck cancer, enlarged surgery causes drastic anatomical and functional changes in the larynx, such as reduced tongue base retraction, cricopharyngeal dysfunction, delayed swallowing reflex, and reduced laryngeal elevation [37,38]. Both reduced elevations of the larynx and delayed swallowing reflex are independent risk factors for aspiration pneumonia in patients with advanced head and neck cancer [39]. Therefore, the risk of pneumonia appears to be high after surgery for head and neck cancer. Thus, the possibility of combined BSIs should be considered in patients with head and neck cancer and pneumonia.

Our study has some limitations. First, the findings of this study may not apply to all patients with head and neck cancer as this was a single-center study. However, it is noteworthy that this is the largest study to date to examine BSIs in patients with head and neck cancer over the last decade. Second, in the non-BSI group, there were cases wherein a specific infectious disease was diagnosed based on clinical symptoms or other test results even when the causative bacteria were not identified by various culture tests. Third, the BSI group is considered to have had a poorer prognosis than the non-BSI group; however, no significant difference was found in the log-rank test. This is considered to be due to the small number of cases in this study. Fourth, performance status is a major factor in the prognosis of tumors and infections, but it was difficult to retrieve this data from the medical chart.

## 5. Conclusions

In summary, we described the biological and epidemiological features of BSIs in patients with head and neck cancer, with CRBSIs being the most frequent source; therefore, CVC or CV port insertion should be performed over a short period of time. Additionally, our study had many cases of BSIs associated with pneumonia; hence, it is necessary to pay attention to the occurrence of BSIs in patients with pneumonia. Understanding the epidemiology of BSIs is important for developing strategies to prevent and control infections. Furthermore, the intervention of the infectious disease department may contribute to the appropriate use of antimicrobial drugs and the prevention of the occurrence of drug-resistant bacteria. Therefore, active participation of the infectious disease department is desirable in BSI treatment. Immediate measures are needed to reduce CRBSIs in patients with head and neck cancer.

## Figures and Tables

**Figure 1 jcm-11-04820-f001:**
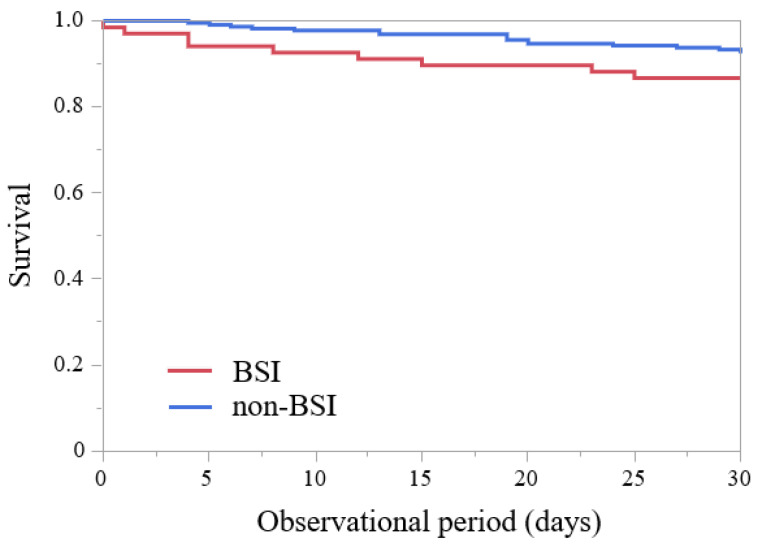
Kaplan-Meier analysis of patients with head and neck cancer. *p* = 0.450 (log rank test). BSI, bloodstream infection.

**Table 1 jcm-11-04820-t001:** Characteristics of bloodstream infections in patients with head and neck cancer.

	Non-BSI Group (*n* = 221)	BSI Group (*n* = 67)	*p*-Value
**Demographic**			
Sex (male, %)	174 (78.7%)	50 (74.6%)	0.504
Age, years, median (IQR)	69 (64–75)	71.0 (64.5–76)	0.367
**Comorbidities**			
Diabetes mellitus	48 (21.7%)	14 (20.0%)	1.000
Dementia	2 (0.9%)	0 (0%)	1.000
Cerebrovascular disease	17 (7.7%)	7 (10.4%)	0.457
Respiratory disease	37 (16.7%)	18 (26.9%)	0.076
Digestive disease	74 (33.5%)	28 (41.8%)	0.244
Kidney and urological disease	34 (15.4%)	11 (16.4%)	0.849
Brain and nervous system disease	4 (1.8%)	0 (0%)	0.576
**Social history**			
Chronic drinker	127 (57.5%) (*n* = 220)	31 (46.3%)	0.123
Smoking	82 (37.1%) (*n* = 220)	25 (37.3%)	1.000
**Vital signs**			
Body weight, kg, median (IQR)	50.3 (43.9–56.1) (*n* = 220)	48.7 (43.4–53.9)	0.385
Body temperature, °C, median (IQR)	38.2 (37.5–38.8) (*n* = 219)	39.1 (38.4–39.6)	<0.001 *
**Laboratory markers**			
WBC count, /μL, median (IQR)	8868.4 (3600–11,900)	7500 (2950–10,700)	0.478
Neutrophil count, /μL, median (IQR)	8000.4 (2862.5–10,447.5) (*n* = 214)	6470 (2815–9245)	0.836
C-reactive protein, mg/dL, median (IQR)	9.3 (3.7–12)	10.0 (4.0–18.1)	0.036 *
**Neutropenia**	11 (5.0%)	5 (7.5%)	0.541
**Mortality**			
30-day mortality	4 (1.8%)	7 (10.4%)	0.004 *
**Duration of hospital stay, days, median (IQR)**	87.0 (56.5–112.8) (*n* = 218)	86.0 (44–113)	0.590
**Use of antibiotics (appropriate)**		54 (80.6%)	
**Ratio of infectious disease departments participating in treatment**		43 (64.2%)	
**Sets of blood cultures**			
One set	24 (10.9%)	12 (17.9%)	0.141
Two sets	195 (88.2%)	55 (82.1%)	0.217
Four sets	2 (0.9%)	0 (0%)	1.000
**Monomicrobial/Polymicrobial**			
Monomicrobial	12 (92.3%)	53 (79.1%)	
Polymicrobial	1 (7.7%)	14 (20.9%)	
**Site of infection**			
Catheter-related infection	15 (6.8%)	26 (38.8%)	<0.001 *
Respiratory tract infection	50 (22.6%)	13 (19.4%)	0.618
Catheter-associated urinary tract infection	6 (2.7%)	4 (6.0%)	0.136
Gastrointestinal infection	3 (1.4%)	3 (4.5%)	0.141
Pyogenic spondylitis	1 (0.5%)	3 (4.5%)	0.040 *
Thrombophlebitis	0 (0%)	3 (4.5%)	0.012 *
IE	2 (0.9%)	1 (1.5%)	0.550
Mucositis	13 (5.9%)	0 (0%)	0.044 *
SSI	4 (1.8%)	0 (0%)	0.576
Tumor-infection	3 (1.4%)	0 (0%)	1.000
Cervical abscess	2 (0.9%)	0 (0%)	1.000
Contamination	13 (5.9%)		
Others	12 (5.4%)	4 (6.0%)	0.770
Unknown	97 (43.9%)	19 (28.4%)	0.033 *
**CVC/CV port**			
CVC presence	28 (12.7%)	23 (34.3%)	<0.001 *
CV port presence	4 (1.8%)	7 (10.4%)	0.004 *
CVC duration before BSIs occurred, days, median (IQR)	21 (12.8–41.3)	20 (11–35)	0.705
CV port duration before BSIs occurred, days, median (IQR)	87 (28.5–204.3)	24 (20–153)	0.412
CVC removal	17 (60.8%)	18 (78.3%)	0.034 *
CV port removal	4 (100%)	5 (71.4%)	<0.001 *
**Tumor site**			
Oral cavity	70 (31.7%)	18 (26.9%)	0.545
Oropharynx	47 (21.3%)	11 (16.4%)	0.487
Hypopharynx	49 (22.2%)	18 (26.9%)	0.415
Larynx	23 (10.4%)	10 (14.9%)	0.380
Carcinoma of maxilla	13 (5.9%)	4 (6.0%)	1.000
Others	19 (8.6%)	6 (9.0%)	1.000
**Cancer stage**	37 (16.7%)	12 (17.9%)	0.853
Early stage (Stage I–II)	167 (75.6%)	51 (76.1%)	1.000
Locally advanced (Stage III–IV)	17 (7.7%)	4 (6.0%)	1.000
Recurrence or metastatic disease	140 (63.3%)	39 (58.2%)	0.474
**Treatment**			
Surgical treatment	39 (17.6%)	16 (23.9%)	0.288
Chemotherapy			
FP	0 (0%)	1 (1.5%)	0.233
TPF	17 (7.7%)	5 (7.5%)	1.000
Adriamycin	1 (0.5%)	1 (1.5%)	0.412
Other regimens	2 (0.9%)	1 (1.5%)	0.550
Radiotherapy	26 (11.8%)	9 (13.4%)	0.675
Chemoradiotherapy			
CDDP-RT	57 (25.8%)	6 (9.0%)	0.004 *
FP-RT	7 (3.2%)	1 (1.5%)	0.686
TPF-RT	4 (1.8%)	2 (3.0%)	0.626
DC-RT	2 (0.9%)	2 (3.0%)	0.232
Biotherapy			
Cmab	2 (0.9%)	0 (0%)	1.000
Cmab-FP	9 (4.1%)	6 (9.0%)	0.124
Cmab-RT	6 (2.7%)	3 (4.5%)	0.439
Nivolumab	2 (0.9%)	1 (1.5%)	0.550
Treatment interval/Palliative care	47 (21.3%)	13 (19.4%)	0.864

The blood test was usually performed on the same day as the blood culture collection. When the collection times for blood testing and blood cultures were not on the same day, the most recent blood test results were adopted. IQR, interquartile range; BSI, bloodstream infection; CVC, central venous catheter; CV port, central venous port; IE, infectious endocarditis; SSI, surgical site infection; FP, fluorouracil + cisplatin; TPF, docetaxel + cisplatin + fluorouracil; CDDP, cisplatin; DC, docetaxel + carboplatin; Cmab, cetuximab; RT, radiation therapy. * *p* < 0.05.

**Table 2 jcm-11-04820-t002:** Factors independently associated with bloodstream infections in patients with head and neck cancer.

Variable	Adjusted OR (95% CI)	*p*-Value
Sex	1.580 (0.759–3.288)	0.221
Age	1.025 (0.992–1.059)	0.137
Respiratory disease	1.787 (0.860–3.717)	0.120
Body temperature, °C	2.563 (1.829–3.593)	<0.001 *
C-reactive protein, mg/dL	1.047 (1.009–1.085)	0.013 *
CDDP-RT	0.336 (0.129–0.870)	0.025 *

OR, odds ratio; CI, confidence interval; CDDP, cisplatin; RT, radiation therapy. * *p* < 0.05.

**Table 3 jcm-11-04820-t003:** Factors independently associated with 30-day mortality due to bloodstream infection in patients with head and neck cancer.

Variable	Adjusted OR (95% CI)	*p*-Value
Sex	3.619 (0.594–22.036)	0.163
Age	0.959 (0.869–1.059)	0.411
Respiratory disease	3.026 (0.486–18.828)	0.235
Body temperature, ℃	0.858 (0.310–2.375)	0.768
C-reactive protein, mg/dL	1.058 (0.967–1.159)	0.220
CDDP-RT	2.695 (0.185–39.271)	0.468

OR, odds ratio; CI, confidence interval; CDDP, Cisplatin; RT, radiation therapy.

**Table 4 jcm-11-04820-t004:** Incidence of causative pathogens.

Causative Organisms	Number	Frequency
Gram-positive coccus		
*Staphylococcus aureus* (MRSA)	18	26.9%
*Staphylococcus epidermidis*	12	17.9%
*Staphylococcus aureus* (MSSA)	4	4.8%
*Enterococcus faecalis*	2	6.0%
*Staphylococcus capitis*	3	4.5%
Others	6	9.0%
Gram-positive bacilli		
*Bacillus cereus*	3	4.5%
*Bacillus* spp.	1	1.5%
Gram-negative bacilli		
*Pseudomonas aeruginosa*	7	10.4%
*Enterobacter aerogenes*	6	9.0%
*Klebsiella pneumoniae*	4	4.8%
*Klebsiella oxytoca*	3	4.5%
*Citrobacter koseri*	2	6.0%
*Enterobacter cloacae*	2	6.0%
Others	4	4.8%
Fungi		
*Candida albicans*	3	4.5%
*Candida parapsilosis*	3	4.5%

MRSA, methicillin-resistant *Staphylococcus aureus*; MSSA, methicillin-susceptible *Staphylococcus aureus.*

## Data Availability

The datasets used and/or analyzed in this study are available from the corresponding author upon reasonable request.

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
