# Peer review of "Clinical and Epidemiological Characteristics of Bloodstream Infections in Head and Neck Cancer Patients: A Decadal Observational Study"

_jcm, 2022, doi:10.3390/jcm11164820_

Round 1

Reviewer 1 Report

General comments

This manuscript conducted a prospective cohort study to assess the BSI in head and neck patients. The authors analyzed 288 cases and classified into non-BSI and BSI group. This manuscript appears to be reporting some significant measurements made on CRBSI. On the other hand, I think that research design and statistics still have some problems as indicated below. Overall, I would suggest extensive revision in combination with re-review for this manuscript.

Specific comments

Major

1)    First of all, epidemiological study means generalizability of some region or country. In this study, the authors only mentioned the patient taken blood culture for examination of BSI. If you mention this study as an epidemiological study, you should describe the population of all head and neck cancer patients in your hospital.

2)    This study area which are the incidence of BSI patients in head and neck cancer are overlapped previous studies such as Uraguchi et al (Supportive care in cancer 2022), Janasen et al. (British journal of caner 2021) and Marin et al (Clin Transl Oncol 2019). What is the difference between a purpose and results of this study and those of these studies?

3)    Enrollment is unclear: are BSI patients only associated with the treatment of head and neck cancer? In my opinion, urinary tract infections can not lead to occur in association with head and neck cancer treatment. Are BSI patients not related to head and neck cancer also considered? It is better to exclude some cases that do not appear to be related to head and neck cancer treatment.

4)    If you set the primary outcome as 90-days mortality, you should insert the figure of the Kaplan–Meier curves and describe the detail of the cause of death. In addition, the outcome of late mortality is unnecessary. Also, please explain why the primary outcome was set at 30- and 90-days, as I believe that most deaths due to BSI occur in the early stage.

5)    You stated, “Valuable with a p-value < 0.10 in the univariable analysis were examined for correlation before inclusion in the multivariate analysis”, and I have serious concerns over these statistics. In this study, since 90-day mortality is used as the primary outcome, gender, age, and performance status, which are necessary for prognosis, should be included as covariates first. The analysis is incorrect because these covariates such as CRBSI and CVC presence have intraclass correlation and CVC removal is probably due to the result of BSI.

6)    Since CRBSI was the most frequent cause of BSIs in patients with head and neck, why not use that as a strong point for deeper discussion? Therefore, please describe the criteria for catheter removal due to CRBSI. In addition, please add the results of catheter tip culture as it is an important diagnostic criterion. If catheter removal was based on culture results, it is not appropriate to list it in Table 1. Also, Non-BSI group cannot be considered CRBSI. Please correct.

Minor

1)    Line 81. Page 2. “The diagnosis of BSIs was defined as at least two sets of positive blood cultures from separate sites: one set positive for a gram-negative bacterial pathogen and one set positive for a gram-positive bacterial pathogen in a patient with an intravascular device and clinical compatibility.” Are you saying that you would not have diagnosed BSI if the blood cultures did not show the same bacteria from separate sites? Could you explain more clearly?

2)    In table 1. “Treatment interval” instead of “treatment interest”.

3)    In table 4. Staphylococcus aureus is duplicated.

Author Response

Responses to Reviewers' Comments

Reviewer 1

Major

  1. First of all, epidemiological study means generalizability of some region or country. In this

study, the authors only mentioned the patient taken blood culture for examination of BSI. If you mention this study as an epidemiological study, you should describe the population of all head and neck cancer patients in your hospital.

Response: We strongly appreciate the reviewer's comments. We have added the number of patients who were hospitalized or underwent outpatient surgery in the Department of Otolaryngology, Head and Neck Surgery during the study period and the percentage of patients who underwent blood culture tests on page 4, line 159, as follows:

“During this study period, a total of 4,958 patients (excluding duplicate patients) were hospitalized or underwent relevant outpatient surgery, and, of these, 288 cases (5.8%) had blood cultures collected.”

  1. This study area which are the incidence of BSI patients in head and neck cancer are overlapped previous studies such as Uraguchi et al (Supportive care in cancer 2022), Janasen et al. (British journal of cancer 2021) and Marin et al (Clin Transl Oncol 2019). What is the difference between a purpose and results of this study and those of these studies?

Response: Thank you very much for this question. In the study, CRBSI was the most common focus of infection of head and neck cancer patients with BSI, followed by pneumonia. In multivariate analyses, the risk factors for BSI were CRBSI, respiratory disease, body temperature, and C-reactive protein, and the risk factors for mortality were C-reactive protein.

We believe it is very important to compare the results with previous studies on the focus and risk factors of these infections and have added such information to the text on page 9, line 256, on page 9, line 269, and on page 10, line 310, as follows:

1) “In a recent study of BSI in patients with head and neck cancers, the 30-day mortality rate was 26%, and BSIs were involved in 10% of early noncancer deaths [12].”

2) “In past studies of significant risk factors for BSI with head and neck cancer, Marin et al. reported hypoalbuminemia, previous chemotherapy, and cetuximab therapy, and Jen-sen et al. reported increasing age, stage, and performance status [12, 14]. In this study, CRBSI, respiratory disease, body temperature, and C-reactive protein were significantly related to the increased risk of BSI. Each study identified different related factors, but the factors lead in common to exacerbations of overall ill condition and de-creased activity.”

3) “In the reports of Marin et al., pneumonia was the most common focus of infection (14 cases, 27.0%), CRBSI was the third most common (eight cases, 16.0%) [14], and the re-sults were roughly similar to those in this study. The results suggest that CRBSI and pneumonia are important as foci of BSI in patients with head and neck cancer.”

Although none of the previous studies presented evaluated the changes in outcomes due to treatment interventions in the Department of Infectious Diseases, we believe that this study is novel in that it has been clarified that intervention from a department of infectious disease promotes the proper use of antibiotics. In replying to this review, we consulted with Dr. Koichi Tokuda, an infectious disease specialist, regarding the diagnosis and treatment of infectious diseases. He has made a great contribution to this paper and is, therefore, now included as a co-author.

  1. Enrollment is unclear: are BSI patients only associated with the treatment of head and neck cancer? In my opinion, urinary tract infections can not lead to occur in association with head and neck cancer treatment. Are BSI patients not related to head and neck cancer also considered? It is better to exclude some cases that do not appear to be related to head and neck cancer treatment.

Response: Thank you very much for your comments. As you pointed out, urinary tract infection is a vague expression. When the medical records were checked, patients with head and neck cancers who had urinary tract infections had urethral catheters inserted for reasons such as hydration during administration of anticancer drugs, difficulty urinating due to disuse caused by long-term hospitalization, and terminal infusion management. Therefore, we decided that the expression “catheter-associated urinary tract infections” was more appropriate than “urinary tract infections” and changed the description in the text.

  1. If you set the primary outcome as 90-days mortality, you should insert the figure of the Kaplan–Meier curves and describe the detail of the cause of death. In addition, the outcome of late mortality is unnecessary. Also, please explain why the primary outcome was set at 30- and 90-days, as I believe that most deaths due to BSI occur in the early stage.

Response: Thank you very much for providing important comments. We have added the figure of the Kaplan–Meier curves and added the sentence on page 4, line 169, as follows:

“Kaplan–-Meier analysis of patients with head and neck cancer is shown in Figure 1.”

When creating the Kaplan–Meier curves, we checked the mortality data again and found the error, so the data in the text and table were corrected accordingly. Like the reviewer, we believe it is important to list the reasons for death, and so we conducted additional research with the medical charts and inserted the relevant information on page 4, line 170, as follows:

“The most common causes of 90-day mortality in BSI patients with head and neck cancers were infectious disease (nine cases, 69.2%), followed by cancer (three cases, 23.1%) and cerebral hemorrhage (one case, 7.7%). In non-BSI patients, the rate of deaths from in-fectious diseases was low (six cases, 20.7%), and the rate of deaths from cancer was relatively high (15 cases, 51.7%). Other causes of death for the non-BSI group were disseminated intravascular coagulation and unknown (two cases each, 6.9%), as well as electrolyte abnormalities, interstitial pneumonia, acute respiratory distress syndrome, and cerebral hemorrhage (one case each, 3.4%). For the BSI group's 90-day mortality rate, out of nine cases where infectious diseases were the cause of death, only five cases (55.6%) had the first-time BSI as the direct cause of death. For the BSI group's early mortality rate, however, infectious diseases were the cause of death in seven cases, and many of the deaths came from the first BSI episode (five cases, 71.4%).”

Based on the above, and in agreement with the reviewer’s comment, BSI was thought to have a greater effect on mortality in the early stages of infection (30 days) than in later stages of infection (30–90 days). Therefore, we have decided that it is not appropriate to include late mortality and have removed the description of late mortality from the text.

  1. You stated, “Valuable with a p-value < 0.10 in the univariable analysis were examined for correlation before inclusion in the multivariate analysis”, and I have serious concerns over these statistics. In this study, since 90-day mortality is used as the primary outcome, gender, age, and performance status, which are necessary for prognosis, should be included as covariates first. The analysis is incorrect because these covariates such as CRBSI and CVC presence have intraclass correlation and CVC removal is probably due to the result of BSI.

Response: Thank you for your invaluable comments. It is quite disappointing that performance status could not be extracted from the database. However, we have added age and sex to the multivariate analysis considerations, and “CVC presence,” “CV port presence,” “CVC removal,” and “CV port removal” have been excluded from the text and Tables 2 and 3.

  1. Since CRBSI was the most frequent cause of BSIs in patients with head and neck, why not use that as a strong point for deeper discussion? Therefore, please describe the criteria for catheter removal due to CRBSI. In addition, please add the results of catheter tip culture as it is an important diagnostic criterion. If catheter removal was based on culture results, it is not appropriate to list it in Table 1. Also, Non-BSI group cannot be considered CRBSI. Please correct.

Response: Thank you very much for these helpful suggestions. With regard to CRBSI cases, we managed CRBSI mainly in the Department of Infectious Diseases. Definitive diagnosis and treatment of CRBSI was carried out in accordance with IDSA guidelines. Catheters were removed in accordance with the IDSA's guidelines, and this information has been added to the text on page 3, line 111, as follows:

“Basically, the CVC or CV port was removed and cultured when the possibility of CRBSI was suspected [8]. If the infecting organism was a CoNS and there was no suspicion of local or metastatic complications, the CVC or CV port was retained.”

Treatment involving catheter removal in the case of CRBSI in the BSI group was per recommendation from the Department of Infectious Diseases to the main department. In the non-BSI group, there were many cases where CRBSI was suspected due to fever, leading to catheter removal based on local departmental judgement only, but the blood cultures proved to be negative.

The results of catheter tip culture have also been added to the text on page 5, line 200, as follows:

“For the BSI group, catheter tip culture was performed in all cases where the catheter was removed. There were 15 cases (65.2%) in which the same bacterial pathogens in the blood culture were detected from the catheter tip. There were five cases (21.7%) where catheter tip culture was negative. Catheter removal followed administration of anti-bacterial drugs and was only after a considerable amount of time had passed in many cases, explaining why bacteria were not detected in some catheter-tip cultures. There were three cases (13.0%) of suspected contamination.”

Also, we completely agree that it is not appropriate to include non-BSI cases suspected of catheter infection in the CRBSI group, so they have been reclassified in Others.

Minor

  1. Line 81. Page 2. “The diagnosis of BSIs was defined as at least two sets of positive blood cultures from separate sites: one set positive for a gram-negative bacterial pathogen and one set positive for a gram-positive bacterial pathogen in a patient with an intravascular device and clinical compatibility.” Are you saying that you would not have diagnosed BSI if the blood cultures did not show the same bacteria from separate sites? Could you explain more clearly?

Response: Thank you very much for your insightful observation. The description pointed out is difficult to understand, so I have corrected it on page 2, line 85, as follows:

“The diagnosis of BSI was defined as one or more blood cultures positive for known pathogenic organisms (e.g., Staphylococcus aureus, gram-negative bacilli, and fungi).”

  1. In table 1. “Treatment interval” instead of “treatment interest”.

Response: Thank you very much for noticing this. We have corrected the description from “treatment interest” to “treatment interval” in Table 1.

  1. In table 4. Staphylococcus aureus is duplicated.

Response: Thank you very much for noticing this as well. The “Staphylococcus aureus” designation was intended to indicate MSSA and MRSA separately, so we have modified the text in Table 4 accordingly.

Reviewer 2 Report

      The manuscript entitled “Clinical and epidemiological characteristics of bloodstream infections in head and neck cancer patients: a decadal observational study” is well written.

         I have only minor comments.

       The authors described the biological and epidemiological features of bloodstream infections in patients with head and neck cancer. The main drawback of this study is that this was a single-centre study conducted at Tohoku University Hospital.

        Authors must discuss more similar studies as they missed several recent literature eg. (PMID: 34017084, PMID: 29948973 etc.)

Author Response

Reviewer 2

Minor

The authors described the biological and epidemiological features of bloodstream infections in patients with head and neck cancer. The main drawback of this study is that this was a single-centre study conducted at Tohoku University Hospital. Authors must discuss more similar studies as they missed several recent literature eg. (PMID: 34017084, PMID: 29948973 etc.)

Response: We strongly appreciate the reviewer's comments. It is, indeed, important to compare the results with previous studies on the focus and risk factors of these infections. We have added relevant information to the text on page 9, line 256, on page 9, line 269, and on page 10, line 310, as follows:

1) “In a recent study of BSI in patients with head and neck cancers, the 30-day mortality rate was 26%, and BSIs were involved in 10% of early noncancer deaths [12].”

2) “In past studies of significant risk factors for BSI with head and neck cancer, Marin et al. reported hypoalbuminemia, previous chemotherapy, and cetuximab therapy, and Jensen et al. reported increasing age, stage, and performance status [12, 14]. In this study, CRBSI, respiratory disease, body temperature, and C-reactive protein were significantly related to the increased risk of BSI. Each study identified different related factors, but the factors lead in common to exacerbations of overall ill condition and de-creased activity.”

3) “In this study, the most frequent source of BSIs in patients with head and neck cancer was CRBSIs, followed by pneumonia. In the reports of Marin et al., pneumonia was the most common focus of infection (14 cases, 27.0%), CRBSI was the third most common (eight cases, 16.0%) [14], and the results were roughly similar to those in this study. The results suggest that CRBSI and pneumonia are important as foci of BSI in patients with head and neck cancer.”

Round 2

Reviewer 1 Report

major

1.     Is the reference you cited, number 12 is about a paper on aspiration pneumonia, or does it describe mortality for BSI? Please confirm.

2.     Please write about Tumor status in terms of T classification, recurrence and metastasis, etc.

3.     Performance status is a major factor in the prognosis of tumors and infections; please explain in limitation.

4.     The mortality between BSI group and non-BSI group was no significant difference at 90-days and 30-days in either case. I think that the mortality rate should be generally higher in the BSI group, therefore, this may be due to small sample size or inappropriate statistics. When it comes to infectious diseases in this article, you should exclude the death except for infection. If you do that, I think there will be a statistically significant difference in this study. In addition, If you focus on infectious diseases, shouldn't we focus on early mortality rate? Also, the log-rank test should be mentioned.

5.     Certainly, there was a statistically significant difference in CRP, but there does not seem to be a clinical importance that would lead to a prediction of BSI group or prognosis.

6.     Would it not be wrong to define CRBSI as a variable in Table 2 or 3 because CRBSI is already a confirmed BSI, unlike variables such as respiratory disease or CRP.

Author Response

Responses to Reviewer's Comments

Reviewer 1

The authors would like to thank the reviewer for their constructive critique to improve the manuscript. We have made every effort to address the issues raised and to respond to all comments. The revisions are indicated in red font in the revised manuscript. Please, find next a detailed, point-by-point response to the reviewer's comments. We hope that our revisions will meet the reviewer’s expectations.

Major

  1. Is the reference you cited, number 12 is about a paper on aspiration pneumonia, or does it describe mortality for BSI? Please confirm.

Response: We would like to thank the reviewer for the insightful comment. When we checked the text and references, we found that references #12 and #32 were reversed. Therefore, we have revised the reference numbers accordingly.

  1. Please write about Tumor status in terms of T classification, recurrence and metastasis, etc.

Response: We would like to thank the reviewer for the constructive comments. As the reviewer pointed out, the T classification is considered to be an important item for understanding the characteristics of bacteremia in patients with head and neck cancers. As the types of head and neck cancers in this study were wide-ranging, it would be complicated to describe each T-classification. Therefore, after referring to a previous research of Uraguchi et al., we have described it as a stage classification based on the TNM classification. Moreover, we have provided more information concerning recurrence and metastasis.

In addition, the definition of the tumor type and stage classification was added to the revised manuscript as follows:

“Histological type and grading of the tumor were evaluated according to the standard International Classification of Diseases for Oncology third edition [12], third edition first revision [13], and third edition second revision [14]. The tumors were staged according to the UICC TNM classification (sixth edition UICC 2002 [15], seventh edition UICC 2010 [16], and eighth edition UICC 2017 [17]).” (Lines 117–121)

Further, the results of the examination on stage classification and the presence or absence of recurrence/metastasis are also presented in Table 1 and in the revised manuscript as follows:

“There was no significant difference in the sites or stage of cancer and types of treatment between the BSI and non-BSI groups.” (Lines 210–212)

  1. Performance status is a major factor in the prognosis of tumors and infections; please explain in limitation.

Response: We would like to thank the reviewer for the comments. As the reviewer pointed out, the performance status is a major factor in the prognosis of tumors and infections. Therefore, we have discussed this issue as a limitation as follows:

“Third, the performance status is a major factor in the prognosis of tumors and infections, but it was difficult to retrieve the data from the medical chart.” (Lines 370–372)

  1. The mortality between BSI group and non-BSI group was no significant difference at 90-days and 30-days in either case. I think that the mortality rate should be generally higher in the BSI group, therefore, this may be due to small sample size or inappropriate statistics. When it comes to infectious diseases in this article, you should exclude the death except for infection. If you do that, I think there will be a statistically significant difference in this study. In addition, if you focus on infectious diseases, shouldn't we focus on early mortality rate? Also, the log-rank test should be mentioned.

Response: We strongly appreciate the reviewer's comments. As the reviewer pointed out, the presence of infectious diseases was the most important reason for death in this study. Moreover, when focusing on infectious disease mortality, we considered that 30-day mortality might have been more important than 90-day mortality. Therefore, we adopted only 30-day mortality and reexamined it, focusing only on deaths owing to infectious diseases.

Please note that we have made the following revisions in the manuscript:

“Interestingly, the 30-day mortality rate was significantly higher in the BSI than in the non-BSI group [seven cases (10.4%) vs. four cases (1.8%), p=0.004]. The outcome of Kaplan–Meier analysis of patients with head and neck cancer is presented in Figure 1.” (Lines 162–165)

“However, the 30-day mortality rate tended to be lower in the group with intervention from the infectious disease department than the group without any such intervention [three (7.0%) vs. four cases (16.7%)].” (Lines 301–303)

The definition of 30-day mortality is also included in revised manuscript:

“Moreover, 30-day mortality was considered as death due to an infectious disease.” (Lines 127)

In addition, we have presented the results of the log rank test as follows:

“The Kaplan–Meier survival estimates over time showed no significant difference between the BSI and non-BSI groups (p = 0.450).” (Lines 165–166)

  1. Certainly, there was a statistically significant difference in CRP, but there does not seem to be a clinical importance that would lead to a prediction of BSI group or prognosis.

Response: We would like to thank the reviewer for the insightful comments. As the reviewer pointed out, CRP does not appear to be of clinical significance to predict BSI group or prognosis. When the 30-day mortality was limited to infections only and the risk factor of death was examined again, CRP did not apply to the risk factor of death. Therefore, we have removed the description from the text.

  1. Would it not be wrong to define CRBSI as a variable in Table 2 or 3 because CRBSI is already a confirmed BSI, unlike variables such as respiratory disease or CRP.

Response: We would like to thank the reviewer for the insightful suggestion. Please note that we have excluded “CRBSI” from the multivariate analysis, and revised Tables 2 and 3.

Round 3

Reviewer 1 Report

The authors made all the suggested changes, and now I think the manuscript is much clearer. However, there are still many things that are wrong and unclear.

1)    Line 254 and 366. This study is not the largest study about BSI, but the largest study about blood culture tests for head and neck cancer. Rephrase the statement about these.

2)    In Table 1, you made some mistakes about unit or ratio for example Diabetes mellitus, WBC and neutrophil. Please confirm those in the tables and text.

3)    As I explained previously, CRBSI is a catheter-related bloodstream infection. Therefore, you should use “catheter-related infection” instead of “CRBSI” in table 1. In addition, you should add catheter-related infection of non-BSI patients. Make sure CRBSI is used correctly in the sentence.

4)    Just to confirm, what is the coverage of Table 3? All patients or BSI patients?

5)    Line 276. You stated, “In this study, CRBSI, respiratory disease, body temperature, C-reactive protein level, and CDDP-RT were significantly related to an increased risk of BSI development.” However, CDDP-RT was seemed to decrease BSI in Table2. Please explain the reason in the discussion.

6)    Line 301. Consulting infectious disease department to manage BSI would be useful, but I don't think the results of this study can show the effectiveness.

7)    Line 325. You should have deleted the 90-day mortality description.

8)    You should mention the early detection and treatment of BSI from this study to reduce mortality.

9)    Figure 1 shows that BSI seems to have a worse prognosis in comparison with non-BIS, as expected, although the log-rank test does not show a significant difference. It may be due to small in number. Please mention this in the Limitation section.

I hope these comments will be helpful.

Author Response

Responses to Reviewer's Comments

Reviewer 1

The authors would like to thank the reviewer for their constructive critique to improve the manuscript. We have made every effort to address the issues raised and to respond to all comments. The revisions are indicated in red font in the revised manuscript. Please, find next a detailed, point-by-point response to the reviewer's comments. We hope that our revisions will meet the reviewer’s expectations.

Major

1)   Line 225 and 3 This study is not the largest study about BSI, but the largest study about blood culture tests for head and neck cancer. Rephrase the statement about these.

Response: We would like to thank the reviewer for the comment. As the reviewer pointed out, this study is not the largest study about BSI, but the largest study about blood culture tests for head and neck cancer. Therefore, we have rephrased the referenced portion of the text, as follows:

“To our best knowledge, this is the largest study that has analyzed the clinical and epidemiological characteristics of blood cultures obtained from patients with head and neck cancer.” (Lines 227–229)

2)   In Table 1, you made some mistakes about unit or ratio for example Diabetes mellitus, WBC and neutrophil. Please confirm those in the tables and text.

Response: We would like to thank the reviewer for the insightful comment. We have modified the WBC and neutrophil unit of Table 1 as you pointed out. Regarding diabetes mellitus, since patients who have been injected with insulin are defined as having diabetes mellitus, the definition has been added in the manuscript as follows;

“Patients receiving insulin injections were defined as having diabetes mellitus.” (Lines 79–80)

3)   As I explained previously, CRBSI is a catheter-related bloodstream infection. Therefore, you should use “catheter-related infection” instead of “CRBSI” in table 1. In addition, you should add catheter-related infection of non-BSI patients. Make sure CRBSI is used correctly in the sentence.

Response: Thank you once again. ‘CRBSI’ in Table 1 has been corrected to ‘catheter-related infection’, and the number of catheter-related infection cases in the non-BSI group has been added to Table 1. Further, since among non-BSI patients, patients suspected with catheter-related infection were included in ‘others’ before the correction, the number of cases of ‘others’ in Table 1 has also been corrected.

4)   Just to confirm, what is the coverage of Table 3? All patients or BSI patients?

Response: We would like to thank the reviewer for the constructive comment. Table 3 shows independent factors for 30-day mortality in patients with blood culture-positive head and neck cancers. For clarity, the title of Table 3 has been changed as follows;

“Table 3. Factors independently associated with 30-day mortality due to bloodstream infection in patients with head and neck cancer”

5)   Line 200. You stated, “In this study, CRBSI, respiratory disease, body temperature, C-reactive protein level, and CDDP-RT were significantly related to an increased risk of BSI development.” However, CDDP-RT was seemed to decrease BSI in Table2. Please explain the reason in the discussion.

Response: We strongly appreciate the reviewer's comments. As the reviewer pointed out, CDDP-RT was an independent factor that reduced BSI. We have modified the manuscript as follows;

“Factors independently associated with BSI in patients with head and neck cancer were body temperature (adjusted odds ratio [aOR], 2.563; 95% CI, 1.829–3.593) and C-reactive protein level (aOR, 1.047; 95% CI, 1.009–1.085). Cisplatin radiation therapy (CDDP-RT) was an independent factor that reduced BSI prevalence (aOR, 0.336; 95% CI, 0.129–0.870).” (Lines 200–204)

6)   Line 277. Consulting infectious disease department to manage BSI would be useful, but I don't think the results of this study can show the effectiveness.

Response: We would like to thank the reviewer for the comments. In this study, there was no significant difference in mortality between the group of patients with BSI who received intervention from the infectious disease department and the group who did not receive the intervention. However, the 30-day mortality rate tended to be lower in the group who received the intervention than in the other group [3 cases (7.0%) vs. 4 cases (16.7%)]. Therefore, we considered that the intervention of the infectious disease department would be beneficial for BSI management. As the reviewer pointed out, the results of this study are not sufficient to show the effectiveness of interventions from the infectious disease department. However, a prior study also showed the usefulness of interventions from the infectious disease department, and we believe that it is an important matter from the viewpoint of in-hospital infection control. Therefore, the content of manuscript has been modified as follows;

“Furthermore, the intervention of the infectious disease department may contribute to the proper use of antimicrobial agents and lead to favorable outcomes in the treatment of BSIs in patients with head and neck cancer. ” (Lines 277–279)

7)   Line 294. You should have deleted the 90-day mortality description.

Response: Thank you once again. We have deleted the 90-day mortality description.

8)   You should mention the early detection and treatment of BSI from this study to reduce mortality.

Response: We strongly appreciate the reviewer's comment. The following has been added to the text:

“The 30-day mortality rate was significantly higher in the BSI group than in the non-BSI group. Therefore, it is considered that death due to BSI is likely to occur in the early stage of infection. Thus, early detection and treatment of BSI may be helpful in reducing mortality from BSI.” (Lines 237–240)

9)    Figure 1 shows that BSI seems to have a worse prognosis in comparison with non-BIS, as expected, although the log-rank test does not show a significant difference. It may be due to small in number. Please mention this in the Limitation section.

Response: As the reviewer pointed out, the lack of a significant difference in the log-rank test is related to the small sample size of the study. Therefore, we have added the following information to the limitation:

“Third, the BSI group is considered to have had a poorer prognosis than the non-BSI group; however, no significant difference was found in the log-rank test. This is considered to be due to the small number of cases in this study.” (Lines 338–341)
